# Ethanol and Medical Psychotropics Co-Consumption in European Countries: Results from a Three-Year Retrospective Study of Forensic Samples in Spain

**DOI:** 10.3390/toxics11010045

**Published:** 2022-12-31

**Authors:** Maira Almeida-González, Luis D. Boada, Guillermo Burillo-Putze, Luis A. Henríquez-Hernández, Octavio P. Luzardo, María P. Quintana-Montesdeoca, Manuel Zumbado

**Affiliations:** 1Institute of Legal Medicine of Las Palmas, 35016 Las Palmas de Gran Canaria, Spain; 2Emergency Department, Hospital Universitario de Canarias, 38320 San Cristóbal de La Laguna, Spain; 3Toxicology Unit, Research Institute of Biomedical and Health Sciences (IUIBS), Universidad de Las Palmas de Gran Canaria, 35016 Las Palmas de Gran Canaria, Spain

**Keywords:** compulsory autopsies, toxicological findings, medical psychotropics, blood alcohol concentrations, polypharmacy

## Abstract

Ethanol and medical psychotropics (MPs) are legal psychoactive substances widely consumed in Western countries that are routinely detected in standard toxicological analyses at compulsory autopsies, and toxicokinetic interactions between these drugs have been described. However, studies assessing the consequences of this co-consumption are scarce in Europe. We performed a retrospective study on toxicological results from compulsory autopsies in Spain. Thirty-five legal psychotropics, including ethanol, were measured in blood samples from 380 decedents to assess the determinants of such co-consumption. MPs were detected in 42.4% of the subjects. Polypharmacy was frequent in our series (25%), being more frequent in men than in women. More than one-third of the decedents had detectable levels of ethanol, and a significant positive association between ethanol levels and age was evident. About one-third of MPs consumers were also co-consumers of ethanol. The simultaneous consumption of ethanol and MPs was higher in men than in women. Blood alcohol concentrations (BAC) were lower in men who consumed MPs. In polypharmacy deaths, there was a significant negative association between the number of MPs consumed and BAC in men (*r* = −0.097; *p* = 0.029). Our results showed a high prevalence of co-consumption of MPs and ethanol in the European population involved in medico-legal issues and suggest that toxicokinetic interactions may be lowering BAC in men. This is a very worrying result, as it could indicate that the legal blood ethanol limits set by legislation would not be appropriate for men on MPs treatment.

## 1. Introduction

Psychotropics are drugs that act on the central nervous system (CNS), producing a number of effects such as sedation, excitation, and/or hallucinations. Among the legal psychotropics, ethanol, on the one hand, and psychotropic medicines, on the other, merit an in-depth study on their forensic implications due to their high consumption in Western populations [1].

Ethanol is a psychoactive and addictive substance legally consumed in Western countries [2]. The World Health Organization (WHO) has estimated that 2100 million regular alcohol consumers worldwide exist, which means that about 43% of the world’s adult population consumes alcohol [3]. Ethanol consumption is considered a major risk factor for traffic accidents and for other causes and is frequently associated with deaths classified as violent because of their accidental, suicidal, or homicidal origin [4]. Moreover, ethanol usually tops the list of psychoactive substances found in postmortem toxicological analyses [5,6]. For these reasons, this legal psychotropic substance plays a key role in medico-legal issues in Western populations.

The group of psychotropic medications comprises benzodiazepines (BZDs), barbiturates, Z-drugs, antidepressants (ADPs), antipsychotics, and medical opiates (MOs). Currently, the abuse of medical psychotropics (MPs) has reached an epidemic level, with more than 16.7 million people in the United States abusing prescription drugs [1]. These drugs can cause transitory or permanent changes in attention, perception, mood, consciousness, and behavior [7,8]. Therefore, their use/misuse may be associated with traffic or occupational accidents, suicides, and homicides [9]. MPs, like ethanol, are frequently detected in standard toxicological analyses at compulsory autopsies [5,6]. Previous studies have shown that a high percentage of compulsory autopsies were positive for MPs in Spain [10].

MPs and ethanol are frequently consumed concomitantly and, in many cases, detected simultaneously in forensic autopsies [11]. Simultaneous consumption of these legal psychotropic drugs is of concern in forensic toxicology, as ethanol may affect the metabolism of MPs (and vice versa). Consequently, the effects of these drugs on the CNS could be modified [12,13,14]. These pharmacokinetic interactions might be associated with several unexpected cognitive and behavioral effects, increasing the risk of consumers being involved in medico-legal issues (including death) [15]. In this scenario, as ethanol can potentiate the depressive/sedative effects produced by MPs, ethanol consumption could be an important (and often underappreciated) factor in drug-related forensic issues, such as unintentional drug-related deaths. Therefore, unintentional drug-related deaths are an ongoing public health problem in Western countries, with ethanol being involved in most cases, usually in combination with MOs, BZDs, or ADPs [16]. This highlights the importance of a pharmacogenetics studies in drug-related deaths, especially in cases of non-overdose of drugs of abuse [17].

Notwithstanding the above, studies analyzing the forensic relevance of the simultaneous use (or abuse) of ethanol and MPs are scarce in European countries. To fill this gap, we conducted the current study to assess the prevalence, effects, and determinants of the simultaneous consumption of ethanol and MPs in Europe, based on toxicological findings from compulsory autopsies over a 3-year period in Spain.

## 2. Materials and Methods

### 2.1. Study Population

This study was conducted in the Outermost European region of the Canary Islands. This Spanish archipelago has a resident population of about 2 million and a visitor population of about 12 million people per year (mostly European tourists from the United Kingdom, Germany, and Sweden). The archipelago is divided into two provinces: Las Palmas and Santa Cruz de Tenerife. The Institute of Legal Medicine of Las Palmas serves a population of 1.1 million inhabitants of the province of Las Palmas [3,9,18].

In the study period (between January 2015 and December 2017) a total of 414 forensic autopsies were performed at the Institute of Legal Medicine of Las Palmas. In two cases the decedents might have received psychotropic treatment prior to death, in two other cases the subjects were under 18 years of age, and in 30 cases toxicological data were partially missing. Consequently, the results of 380 forensic autopsies performed on subjects over 18 years of age were finally included in the present study.

### 2.2. Toxicological Analysis

Quantitative determination of psychoactive drugs in blood samples was performed. At least 1 mL of blood was taken from the femoral vein at the time of autopsy. When this was not possible, the blood was taken directly from the heart. The availability of the sample depended on the decision of the coroner and the circumstances of the autopsy. In general, the bodies were in a good state of preservation and kept refrigerated until the time of autopsy. Blood samples were stored at −7 °C until analysis. All samples were analyzed less than 4 weeks after recovery. Socio-demographic characteristics of the subjects—age, gender, date, and cause of death—were obtained from forensic reports and studied in relation to the toxicological analyses.

The legal drugs analyzed were (a) ethanol; (b) BZDs and metabolites: midazolam, alprazolam, bromazepam, clonazepam, lorazepam, oxazepam, temazepam, lormetazepam, nordiazepam, chlordiazepoxide, diazepam, and flurazepam; (c) Z-drugs: zolpidem and zoplicone; (d) ADPs: amitriptyline, nortriptyline, maprotiline, paroxetine, sertraline, and citalopram; (e) MOs: morphine, tramadol, codeine, fentanyl, and methadone; (f) antipsychotics: haloperidol, olanzapine, quetiapine, and risperidone; and (g) barbiturates: pentobarbital, phenobarbital, thiopental, secobarbital, and amobarbital.

The analytical method for medical psychotropics has been previously reported [9,19]. Briefly, after a solid-phase extraction procedure, blood sample extracts were analyzed to identify and quantify the investigated substances using a UHPLC model 1290 (Agilent Technologies, Palo Alto, CA, USA), interfaced to an Agilent 6460 Triple quadrupole mass spectrometer, equipped with a jet stream electrospray interface operating in positive ionization mode (Agilent Technologies).

Similarly, the analytical method for ethanol has also been previously reported [3,18]. In brief, samples were analyzed in a Trace-Focus GC headspace gas chromatography system (GC-HS) equipped with a flame ionization detector (FID) (Thermo Fisher Inc., Waltham, MA, USA). This is the preferred method in forensic sciences for determining ethanol and other volatiles in body fluids.

### 2.3. Statistical Analyses

Descriptive statistics (means and standard deviations, medians and 25th and 75th percentiles of the distribution) were calculated for the drugs measured. The Kolmogorov–Smirnoff and Shapiro–Wilk tests were used to verify the normality of the numerical data. Comparison of variables from two independent samples was performed using Student’s t-test (if the data were normally distributed) or the Mann–Whitney U-test (otherwise). To compare variables with non-normal distribution between more than two independent samples, the Kruskal–Wallis test was used. To analyze the association between two numerical variables, Pearson’s linear correlation coefficient was used. Categorical variables were summarized using absolute frequencies and percentages. Equality of proportions of the categories was analyzed using the binomial test and the Chi-Square test (goodness-of-fit). Analysis of the association between two categorical variables was performed using the Chi-square test or Fisher’s exact test. Results were considered statistically significant if *p* < 0.05. Database management and statistical analyses were performed using IBM SPSS Statistics v 27.0 software (IBM Co., New York, NY, USA).

## 3. Results and Discussion

We conducted a retrospective observational study analyzing toxicological data from blood samples collected from 380 compulsory forensic autopsies of subjects over 18 years of age performed during the years 2015 to 2017 at the Institute of Legal Medicine of Las Palmas (Spain). Our findings and a thorough review of the scientific literature on this matter provide important new insights on the prevalence, determinants, and potential interactions between alcohol and psychoactive medications, taken alone or in combination, in a European adult population involved in medico-legal issues.

As shown in Table 1, among the autopsied subjects included in this study (n = 380), 285 (75%) were male, and 95 (25%) were female. Most of the subjects were over 45 years, and the mean age of the decedents was 49.4 years old (with a range between 18 and 82 years). Among them, 103 individuals were victims of violent death, and 42 were victims of traffic accidents. The cases were evenly spread over the three years included in the study: 2015, 2016, and 2017. Table 1 summarizes the demographic and forensic characteristics of the study population.

It is striking that more than 60% of the subjects (66%) tested positive for at least one legal psychotropic drug. The frequency of detection, the concentrations (mean and standard deviation and median (and interquartile range)), and limits of detection (LODs) and quantification (LOQs) of drugs listed above are shown in Table 2.

Our results demonstrate that legal psychotropics are frequently found in compulsory autopsies of adult subjects in Spain (see Table 2). These findings agree with those previously published, which indicate that BZDs, opiates, and ethanol are the most frequently detected drugs in forensic toxicological analyses in European [10,11] and non-European countries [5,6]. Thus, in our series, residues of MPs were present in more than 40% of the deceased, with BZDs being the most frequently detected psychotropic medicines (25.3% of the study subjects) followed by MOs and ADPs. As shown in Figure 1, a significant association was found between violent death and the frequency of detection of ADPs.

Our results, which show that more than 40% of the study subjects had detectable residues of MPs, are particularly worrying. However, it should not be forgotten that the population in Spain shows a high rate of consumption of these drugs [20,21,22]. The Organization for Economic Co-operation and Development (OECD 2015–2017) reported that the volume of sales of antidepressants and anxiolytics in outpatient services was higher in Spain compared with the rest of the 25 OECD countries [23]. In this sense, our findings seem to reflect the pattern of consumption of these drugs by the Spanish population. In this context, it should be noted that toxicological detection of prescribed medicines in forensic postmortem studies could be a useful indicator of ongoing pharmacotherapy in a region [24,25]. However, the problematic use of psychotropic drugs has risen drastically in developed countries. Due to their effects on the CNS, an increased presence and involvement of medical psychotropics in forensic issues seems evident [26,27]. Thus, in this same series, our group reported that BZDs, MOs, and ADPs were found more frequently in men who suffered a violent death and that BZDs use was positively associated with violent death [10], similar to data reported in Australia, where BZDs were detected in 13.2% of homicide and suicide deaths [28,29].

As shown in Table 3, 51 subjects demonstrated polypharmacy (consumption of three or more MPs simultaneously), with this pattern of consumption being more frequent in men compared to women. The existence of polypharmacy, defined as the use of several psychotropic drugs simultaneously [28], may be of concern due to the potential occurrence of drug–drug interactions and unexpected toxicity phenomena [29,30]. Although polypharmacy is known to increase the risk of road traffic accidents [27,28,29,30,31], our findings showed an absence of association between traffic victims and polypharmacy, suggesting that road traffic accident victims were rarely associated with polypharmacy in our study (Figure 1).

On the other hand, as described in Table 2, more than 35% of the subjects (136) tested positive for ethanol. As shown in Figure 2, a positive correlation was evident between blood alcohol concentration (BAC) and age in the whole series (*r* = 0.237; *p* = 0.007). It should be highlighted that among ethanol consumers, 47 were also co-consumers of MPs. Simultaneous consumption of ethanol and MPs was more frequent in men compared to women (Figure 1). Notably, the median BAC in ethanol and MPs co-consumers was lower in men than in women. Again, the absence of a relationship between traffic accidents and ethanol levels was evident in subjects with co-consumption of MPs (see Table 3). However, as shown in Table 3, there was a significant relationship between violent death and BAC as compared to all other causes of death. These results are consistent with those published previously in other European countries, which report relatively similar detection frequencies of alcohol and generally higher prevalence of alcohol in violent deaths [32,33,34].

It was striking that 17 subjects under polypharmacy were also co-consumers of ethanol. In these subjects, lower BAC was evident in males compared to women, and higher BAC was related to violent death (Table 3). Considerable changes in the pharmacokinetics of MPs are expected when ingested together with alcohol. The simultaneous consumption of ethanol and MPs is a key issue of relevance in forensic toxicology, since ethanol can produce additive or synergistic effects with other CNS depressants, potentiating the effects of these psychotropics [11]. These alterations can lead to the damage of many motor, sensory, and neurological functions, and even sudden death [11,13,34].

Although there is a possibility that postmortem redistribution may affect the blood MPs levels, it must be said that MPs measured in our study show a low postmortem redistribution, so the precision of forensic toxicological findings is high [10]. Regarding ethanol, postmortem ethanol diffusion is a debated issue, with significant difference between heart and peripheral blood concentration [35]. Nevertheless, the existence of intra-individual variations (redistribution) does not play a relevant role in the results and conclusions of our retrospective epidemiological study because we compare inter-individual levels. 

Concurrent use of alcohol and some prescription drugs has been reported to be associated with several cognitive and behavioral effects, which increase the risk of accidents or other severe problems, with legal consequences [14,27]. In this regard, ethanol was a co-intoxicant in 18.5% of opioid-related deaths and 21.4% of benzodiazepine-related deaths in 2010, and in 11% of synthetic opioid-related deaths in 2016 [16]. However, how this combination of legal psychotropics causes toxicokinetic alterations is not well known, and its exact mechanism remains unclear. Perhaps one of the main problems is the unpredictability of the effect of alcohol on drug pharmacokinetics, given the variable nature (and habits) of alcohol use in most subjects. In this scenario, it is noteworthy that it has been reported that co-ingestion of alcohol and psychotropic drugs is associated with a significant decrease in opiate levels in drug deaths, indicating that blood MPs concentration is lower when alcohol is present [14,16,36,37]. Conversely, the presence of MPs seems to induce a decrease in BAC values, as reported by Koski et al. (2003) in Finnish autopsy samples [38]. In fact, our findings suggesting a decrease in BAC in men in the presence of ADPs (Table 3) agree with findings describing a decrease in BAC in the presence of tricyclic antidepressants in a Finnish population [38]. In addition, our results showing a decrease in BAC in men co-consuming MPs (Table 3) and the existence of a negative correlation between BAC and the number of MPs ingested (Figure 3) reinforce the results reported by Koski et al. (2003) in the sense that the presence of MPs can induce a decrease of blood ethanol levels in men [38]. 

Considering this, although for alcohol metabolism the hepatic alcohol dehydrogenase (ADH) is the main enzyme responsible for this process, for other enzymes, the microsomal mixed-function oxidases (specifically CYP2E1) are involved. Interestingly, this microsomal system is also involved in the metabolization processes of several MPs, such as BZDs, ADPs, and MOs [11,35,37,38]. Although the mechanism leading to these toxicokinetic interactions is not well defined, toxicokinetic interactions may occur during absorption, distribution, and excretion, and not only during metabolization processes [11]. In any case, our findings suggesting an inverse association between BAC and MPs show an evident sexual dimorphism, mainly affecting males. This very striking result could probably be related to existing toxicokinetic differences between sexes. Thus, there is a possibility that these sex differences are due to differences in alcohol dehydrogenase isozyme activity [38]. In any case, other factors, such as inter-individual differences, have to be taken into account because inter-individual variations in the DNA sequence may affect the pharmacokinetics or pharmacodynamics of drugs (pharmacogenetics) [17,39]. In this context, the lack of pharmacogenetic information on the decedents constitutes a limitation of our study, although it does not invalidate our conclusions, since the sample size is sufficiently large to support them.

Furthermore, current evidence suggests that women have the same rate of alcohol elimination (i.e., eliminate approximately the same amount of alcohol per unit body weight per hour) but a higher alcohol disappearance rate (i.e., they remove more alcohol per unit of lean body mass per hour) compared to men [40]. All these differences, together with the fact that women have a lower proportion of body water than men of similar body weight and therefore achieve a higher BAC after consuming equivalent amounts of alcohol [41], may explain these sex differences.

However, in conclusion, it could be said that the interactions between ethanol and psychotropic drugs are evident and worrying because of their effects on the CNS. As mentioned above, whatever the underlying toxicokinetic mechanism, our results provide relevant insight into the interactions between MPs and ethanol, with implications for drug safety and forensic toxicology; in cases of co-consumption with ethanol and MPs, ethanol levels might not be realistic. The potential CNS-induced effects of ethanol and MPs may be underestimated, at least in men. Although more information (and research) is needed, the possibility that legal blood ethanol limits set by legislation (for vehicle drivers, for example) may not be adequate in the case of men on MPs treatment is a matter of great concern given the widespread use of psychotropics, ethanol, and MPs in Western countries, especially in higher-income countries where the increasing use of BZDs and ADPs is particularly pronounced, even as a mode of suicide or self-poisoning [42,43]. Community-based studies are needed to assess the pharmacological, toxicological, or forensic implications of potential alcohol–drug interactions.

## Figures and Tables

**Figure 1 toxics-11-00045-f001:**
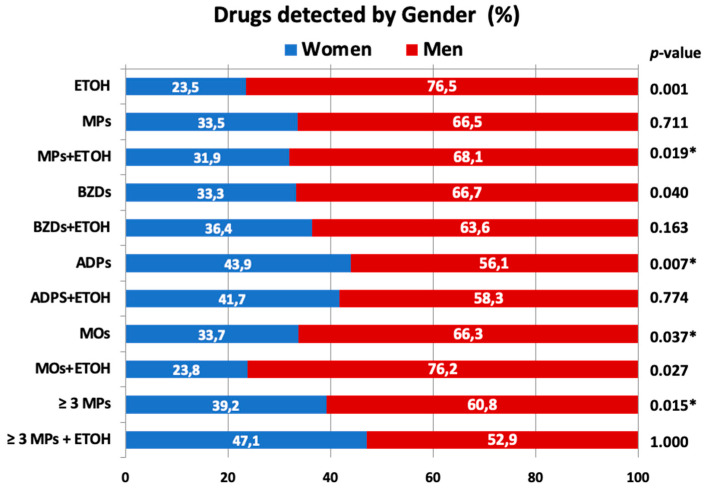
Frequencies (percentages) of group of drugs detected in relation to sociodemographic (top) and forensic characteristics (bottom). ETOH: ethanol; MPs: medical psycothropics; BZDs: benzodiazepines; ADPs: antidepressants; MOs: medical opiates. The asterisk indicates the existence of statistical significance.

**Figure 2 toxics-11-00045-f002:**
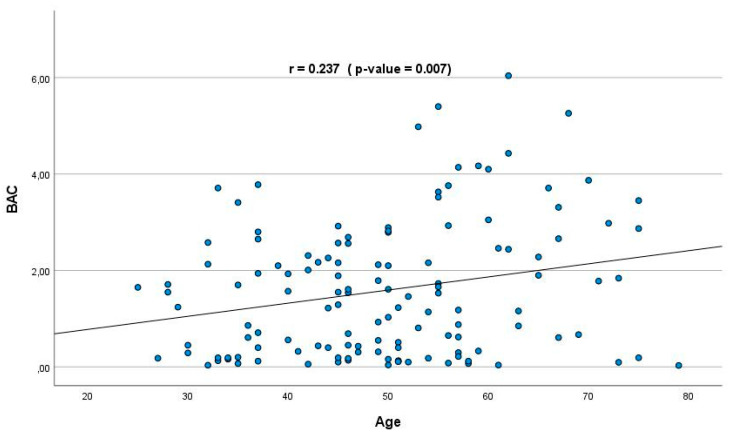
Positive association in whole series among age and blood alcohol concentrations (BAC; g/L). *r* = 0.237; *p* = 0.007; Pearson’s linear correlation coefficient.

**Figure 3 toxics-11-00045-f003:**
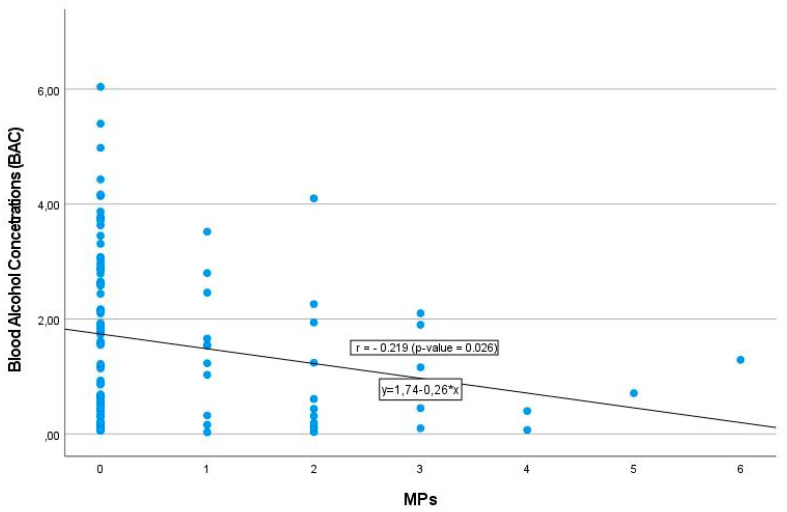
Inverse relationship in men among number of MPs detected and blood alcohol concentrations (BAC; g/L). *r* = 0.219; *p* = 0.026, Pearson’s linear correlation coefficient.

**Table 1 toxics-11-00045-t001:** Description of subjects by sociodemographic characteristics and forensic data.

Category		n (%)
Whole series		380 (100%)
Date (year)		
	2015	121 (31.8%)
	2016	103 (27.1%)
	2017	156 (41.1%)
Gender		
	Male	285 (75.0%)
	Female	95 (25.0%)
Age (years)		
	≤45	129 (33.9%)
	>45	238 (62.5%)
Cause of death		
	Traffic accident	42 (11.1%)
	Violent death ^a^	103 (27.1%)
	Suicide	81 (21.3%)
	Other ^b^	154 (40.5%)

^a^ Includes firearms, drowning, falls, and overdose. ^b^ Include heart disease, infection, gastrointestinal disease, sudden death, and unknown causes.

**Table 2 toxics-11-00045-t002:** Mean (and Standard Deviation) and Median (and Interquartile Range: Percentile75–Percentile25) in ng/mL (except for ethanol, g/L), limit of detection (LOD), limit of quantitation (LOQ) in ppb (ng/mL or g/L for ethanol), and frequency of detection of analyzed compounds.

Group	Compound	n (%)	Mean (SD)	Median (IR)	LOD	LOQ
Benzodiazepines (BZDs)		96 (25.3)				
	Midazolam	17 (4.5)	367.6 (455.9)	180.6 (621.4)	1.50	4.50
	Alprazolam	18 (4.7)	226.7 (201.4)	163.7 (254.9)	1.50	4.50
	Bromazepam	0	-	-	6.25	18.75
	Clonazepam	0	-	-	6.25	18.75
	Lorazepam	6 (1.6)	61.5 (33)	57.1 (63.8)	6.25	18.75
	Oxazepam	15 (3.9)	279.5 (288.5)	183.1 (297.2)	6.25	18.75
	Temazepam	16 (4.2)	534.0 (654.8)	258.2 (484.0)	1.50	4.50
	Lormetazepam	6 (1.6)	273.2 (396.5)	112.8 (382.4)	1.75	6.25
	Nordiazepam	55 (14.5)	1566.2 (3004.2)	600.0 (1561.7)	1.50	4.50
	Clordiazepoxide	1 (0.3)	43.34 (-)	-	1.50	4.50
	Diazepam	35 (9.2)	288.0 (318.7)	160.1 (289.1)	1.00	3.00
	Flurazepam	1 (0.3)	459.0 (-)	-	1.50	4.50
Z-Drugs		7 (1.8)				
	Zolpidem	5 (1.3)	110.9 (228.0)	7.9 (265.5)	1.00	3.00
	Zoplicone	2 (0.5)	180.5 (223.5)	180.5 (-)	6.25	18.75
Antidepressants (ADPs)		41 (10.8)				
	Amitriptiline	8 (2.1)	821.0 (769.9)	705.2 (1501.3)	1.00	3.00
	Nortriptiline	9 (2.4)	265.0 (308.3)	193.7 (400.4)	1.00	3.00
	Maprotiline	6 (1.6)	667.8 (789.9)	366.0 (1504.2)	1.00	3.00
	Paroxetine	12 (3.2)	350.9 (514.8)	143.7 (333.3)	6.25	18.75
	Sertraline	8 (2.1)	399.3 (383.2)	249.3 (571.1)	1.00	3.00
	Citalopram	15 (3.9)	232.5 (148.8)	223.8 (184.6)	1.00	3.00
Medical Opiates (MOs)		90 (24.2)				
	Morphine	33 (8.7)	143.0 (215.0)	71.6 (88.6)	1.00	3.00
	Tramadol	32 (8.4)	1132.3 (3395.9)	174.6 (614.8)	1.00	3.00
	Codeine	7 (1.8)	118.2 (155.7)	23.1 (188.6)	1.00	3.00
	Fentanyl	8 (2.1)	20.5 (36.8)	5.1 (22.7)	0.50	1.50
	Methadone	29 (7.6)	459.3 (474.0)	327.1 (379.0)	1.00	3.00
Antipsychotics		16 (4.2)				
	Haloperidol	1 (0.3)	16.8 (-)		1.0	3.00
	Olanzapine	0	-	-	1.5	4.50
	Quetiapine	15 (3.9)	163.9 (184.0)	-	1.5	4.50
	Risperidone	1 (0.3)	28.0 (-)	-	1.0	3.00
Barbiturates		1 (0.3)				
	Phenobarbital	1 (0.3)	352.5 (-)		1.00	3.00
Ethanol (g/L)		136 (35.8)	1.6 (1.4)	1.5 (2.3)	0.01	0.05

**Table 3 toxics-11-00045-t003:** Median (and interquartile range) of blood alcohol concentrations (BAC) in g/L in relation to type of legal psychotropics detected and sociodemographic and forensic characteristics.

Drugs Detected	Gender			Cause of Death				
	Women	Men	*p* ***	Traffic Accident	Violent Death	Suicide	Other	*p ***
ETOH (n = 136)	1.71 (2.25) (n = 32)	1.23 (2.29) (n = 104)	0.445	1.58 (2.07) (n = 18)	2.13 (2.23) (n = 44)	0.70 (1.60) (n = 30)	0.84 (2.30) (n = 44)	0.009
MPs + ETOH (n = 47)	1.7 (1.46) (n = 15)	0.87 (1.67) (n = 32)	0.057	1.14 (1.19) (n = 4)	2.13 (1.0) (n = 16)	0.32 (1.38) (n = 14)	0.71 (1.29) (n = 13)	0.007
BZDs + ETOH (n = 33)	1.86 (1.59) (n = 12)	0.71 (1.85) (n = 21)	0.089	1.03 (-) (n = 3)	2.13 (1.02) (n = 12)	0.19 (1.77) (n = 9)	0.71 (1.05) (n = 9)	0.060
ADPs + ETOH (n = 12)	2.01 (1.72) (n = 5)	0.46 (1.04) (n = 7)	0.042	-	1.96 (1.97) (n = 6)	0.45 (1.20) (n = 5)	0.31 (-) (n = 1)	0.208
MOs + ETOH (n = 21)	2.16 (2.37) (n = 5)	0.66 (1.33) (n = 16)	0.117	1.39 (-) (n = 2)	2.46 (1.47) (n = 5)	0.29 (0.92) (n = 6)	0.57 (1.35) (n = 8)	0.039
≥3 MPs + ETOH (n = 17)	1.93 (1.31) (n = 8)	0.71 (1.35) (n = 9)	0.012	-	2.16 (0.92) (n = 7)	0.65 (1.14) (n = 6)	1.00 (0.99) (n = 4)	0.003

***** Non parametric Mann–Whitney U test; ****** Non parametric Kruskal–Wallis test.

## Data Availability

Not applicable.

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
