# Peer review of "Ethanol and Medical Psychotropics Co-Consumption in European Countries: Results from a Three-Year Retrospective Study of Forensic Samples in Spain"

_toxics, 2022, doi:10.3390/toxics11010045_

Round 1

Reviewer 1 Report

he paper entitled "Ethanol and Medical Psychotropics co-consumption in European Countries. Results from a Three-Year Retrospective Study in Forensic samples in Spain" deals with the evaluation of the results obtained after the analysis of diferent MPs substances and ethanol in blood samples from decedents in a region of Spain. 

The topic is of interest, and the authors have demonstrated that toxicokinetic interactions between the consumption of different psycotropic substances can have an effect on the ethanol in blood, and regarding this topic there are scarce data in European countries. So, there is an interest for the scientific community in studies as the presented since they evidence that the ethanol levels determined in blood samples from polyconsumers of different types of MP substances are not realistic and can be understimated.

However, from my point of view there are some points that I believe that the authors should explain. First, I suggest that the authors introduce more information about the analytical methodology used in their research, even that they are based on a previous paper, it is still no clear with is the SPE sorbent used for example. In this regard, it could be interesting that the authors also explain why they include in the manuscript some information in table 2 that is redundant with information given in another paper of the same authors (reference 9).

Another point of concern is related with the specific interactions of the MPs with ethanol. The authors explain this in a rather vague way and could be of interest to give more detailed information related to that issue.

Author Response

Reviewer 1

COMMENT: The paper entitled "Ethanol and Medical Psychotropics co-consumption in European Countries. Results from a Three-Year Retrospective Study in Forensic samples in Spain" deals with the evaluation of the results obtained after the analysis of different MPs substances and ethanol in blood samples from decedents in a region of Spain.

The topic is of interest, and the authors have demonstrated that toxicokinetic interactions between the consumption of different psycotropic substances can have an effect on the ethanol in blood, and regarding this topic there are scarce data in European countries. So, there is an interest for the scientific community in studies as the presented since they evidence that the ethanol levels determined in blood samples from polyconsumers of different types of MP substances are not realistic and can be understimated.

ANSWER: We are grateful for the kind words of the reviewer, since there are truly few studies of this type carried out in the European population and it is not a minor issue that the simultaneous intake of alcohol and medical psychotropics could modify the levels of ethanol in blood, especially in men.

COMMENT: However, from my point of view there are some points that I believe that the authors should explain. First, I suggest that the authors introduce more information about the analytical methodology used in their research, even that they are based on a previous paper, it is still no clear with is the SPE sorbent used for example.

ANSWER: We thank the reviewer for his interesting comment. Firstly, we would like to comment that more emphasis was not placed on the analytical method used to avoid unnecessarily lengthening the manuscript. Even more so, because this method was recently published in a previous article. In any case, and following the reviewer's suggestion, we have included a new reference in the text (Hansen et al., For Sci Int 2021) that allows the reader to obtain more details of the method used, such as the type of solid-phase extraction used in the sample preparation prior to toxicological analyses, in this case “pre-conditioned SPE columns (Strata X-C 30 mg/well, Phenomenex, Torrance, CA, USA)”.

Thus, such reference has been introduced in the new version of the manuscript with the number 19, in the line 114, Material and Methods section, and, accordingly, in the line 337 in References section:

“Hansen, S.L.; Klose, M.K.; Linnet, N.K.; Rasmussen, B.S. Simple implementation of muscle tissue into routine workflow of blood analysis in forensic cases – A validated method for quantification of 29 drugs in postmortem blood and muscle samples by UHPLC–MS/MS. For Sci Int 2021, 325: 11090.”

COMMENT: In this regard, it could be interesting that the authors also explain why they include in the manuscript some information in table 2 that is redundant with information given in another paper of the same authors (reference 9).

ANSWER: Although it is possible that the information contained in Table 2 may be considered somewhat redundant with the information shown in a previous article (Almeida-González et al., Toxics 2022, 10(2), 64), what is beyond doubt is that in any forensic toxicology article in which the results, discussion and conclusions are based on the presence or quantification of drugs in biological fluids, it is necessary to show the reader the LODs and LOQs of the different analytes.

For this reason, and greatly appreciating the reviewer's comment, in our opinion it is necessary to keep this table in the present manuscript.

COMMENT: Another point of concern is related with the specific interactions of the MPs with ethanol. The authors explain this in a rather vague way and could be of interest to give more detailed information related to that issue.

ANSWER: We fully agree with the comment made by the reviewer. In fact, the absence of studies about the potential pharmacokinetic (and even pharmacodynamic) interactions of MPs when ethanol is consumed simultaneously with such medical drugs was one of the reasons that led us to conduct this study. The possibility that ethanol may modify the pharmacokinetics and pharmacodynamics of MPs (and vice versa) is of concern in terms of liability for use or misuse.

Precisely because of this lack of information, we decided to focus our discussion on the articles published by Koski et al. in 2003 and 2005, which already described significant variations in plasma levels of psychotropic substances and ethanol when ingested simultaneously. In our opinion, these works carried out on forensic samples from Finland have not been given the attention they deserve, since they describe that “MPs seems to induce a decrease in BAC values”. In any case, one of the main findings of our study reinforce the results described by Koski et al. in 2003. Unfortunately, due to the absence of toxicokinetic studies we could not discuss such interesting finding more deeply. In our opinion, new toxicological studies (with animal models mainly, or epidemiological studies) are necessary to clarify this relevant result.

Reviewer 2 Report

Ethanol and Medical Psychotropics co-consumption in European Countries is important problem, so another study in this area is valuable.

Here are some suggestions:

-        in line 44 please add appropriate citation

-        please discuss more Your finding, that polypharmacy was more frequent in men than in women (what is the reason?)

-        remember that the population of the Canary Islands is quite specific because of the visitor population (ethanol consumption is probably higher in comparison to residents – if there are such comparative studies include it!)

-        did You use heart blood for the determination of MPs? was the possibility of postmortem redistribution taken into consideration during interpretation of the results? write about this phenomenon

Author Response

Reviewer 2

COMMENT: Ethanol and Medical Psychotropics co-consumption in European Countries is important problem, so another study in this area is valuable.

ANSWER: We are grateful for these words of the reviewer. As we answer to the first reviewer, there are a few studies of this type carried out in the European population. Having into account that it is not a minor issue that co-consumption of alcohol and medical psychotropics may modify the levels of ethanol in blood, especially in men, the toxicokinetic interactions between these drugs need to be explored in depth.

COMMENTS: Here are some suggestions:

- in line 44 please add appropriate citation.

ANSWER: Following the appropriate suggestion of the referee, the reference by McHugh et al. (2015) previously mentioned with number 6, has been introduced in the line 44 and renumbered with the number [1], and this change has also been performed in the line 296 in References Section.

McHugh, R.K.; Nielsen, S.; Weiss, R.D. Prescription drug abuse: from epidemiology to public policy. J Subst Abuse Treat 2015, 48, 1-7.

- please discuss more Your finding, that polypharmacy was more frequent in men than in women (what is the reason?).

ANSWER: Although, as appropriately indicated by the referee, this finding might be interesting, the fact that in our series most decedents were men (75%) oblige us to take this result with caution. Due to such circumstances, we did not discuss gender-related pharmacoepidemiological differences in depth throughout this work. Future studies including more subjects are guaranteed to explore this interesting result.

- remember that the population of the Canary Islands is quite specific because of the visitor population (ethanol consumption is probably higher in comparison to residents – if there are such comparative studies include it!).

ANSWER: This is a very appropriate comment of the reviewer. Firstly, we have to say that there are not studies comparing ethanol consumption in the Canary Islands with ethanol consumption in mainland Europe. Although it is true that the Institute of Legal Medicine of Las Palmas serves a very specific region of the Spanish territory (Province of Las Palmas of the Archipelago of the Canary Islands), we believe that our data are of enormous interest precisely because it is an area of high tourist importance for all European countries (the Canary Islands have about 2 million inhabitants and receive 12 million tourists annually, the vast majority of whom are citizens of the European Union). As a consequence, all the subjects included in the study were citizens of the European Union, so that our results reflect (albeit indirectly) what may be happening in the areas of Europe that are under European Union health (and drug) legislation. Therefore, in this case, the fact that it is a highly touristic area adds a plus of interest to the work carried out as it facilitates the extrapolation of results to the rest of Spain, and even to the European Union.

- did You use heart blood for the determination of MPs? was the possibility of postmortem redistribution taken into consideration during interpretation of the results? write about this phenomenon.

ANSWER: As noted by the reviewer, we must take into consideration postmortem redistribution because postmortem redistribution can cause an artificial increase/decrease in postmortem blood concentrations. Thus, in drugs with low postmortem redistribution, the precision of forensic toxicological findings may be high. This seems to be the case for the BZDs, MOs and antipsychotics measured in our study, which show a low postmortem redistribution as reported in the literature (Almeida-González et al., 2022). Therefore, our results showing toxicokinetics interactions between MPs and ethanol may be considered as valid as the blood samples were taken directly from the heart. Nevertheless, as in this work we are evaluating differences in ethanol levels between group of individuals who have ingested psychotropic drugs and whose bodies have remained in the same conditions (under refrigeration), the existence of intra-individual variations (redistribution) does not play a relevant role in the results and conclusions of our study.

In any case, and Following the appropriate suggestion of the referee, a new paragraph about this topic has been added in the new version of the manuscript (Results and Discussion Section, lines 228-235), and a new reference about this topic has also been introduced (Results and Discussion Section, lines 228-235, and in References Section line 281). The new paragraph is the following: “As BZDs, MOs, and antipsychotics measured in our study show a low postmortem redistribution, the precision of forensic toxicological findings is high (Almeida-González et al., 2022). Regarding ethanol, postmortem ethanol diffusion is a debated issue with significant difference between heart and peripheral blood concentration, but the existence of intra-individual variations (redistribution) does not play a relevant role in the results and conclusions of our study because we are comparing inter-individual levels.” And the new reference is:

Martia, V.; Augsburgerb, M.; Widmerb, C.; Lardi, C. Significant postmortem diffusion of ethanol: A case report. For Sci Int 2021, 328, 111046.

Reviewer 3 Report

The manuscript is an interesting article on deaths related to the use of ethanol and psychotropic substances. The article is very interesting however there are some points that need to be clarified in the methods, results, and discussion, in order to proceed with the acceptance of the article which today appears confusing.

1) Methods: what PMI (post-mortem interval) did the deceased have? Were there any subjects with putrefactive phenomena? Were the corpses kept in cells? This aspect is very important because in the stages of putrefaction, with autolysis, the increase of ethanol in the blood has been recorded and this could be an artifact.

2) Methods: how was the toxicological examination done? Was the method unique or did it vary? What kind of method was used?

3) A different paragraph should be made for results and for discussion.

The results are confusing, they must be exposed uncritically and the tables in the text must be explained.

4) Results: why were sudden deaths and gastrointestinal disorders inserted in line 156? What does that have to do with the study?

3) Discussion:

The authors should better explain the purpose of the study. Why should the scientific community be interested? Was the cause of death related to the study of these substances acutely or chronically?

6) it is necessary to cite these studies in the present study:

https://doi.org/10.3390/toxics9110292

Wolf, C.R.; Smith, G.; Smith, R.L. Pharmacogenetics. BMJ 2000, 320, 987–990.

Thieme, D.; Rolf, B.; Sachs, H.; Schmid, D. Correlation of inter-individual variations of amitriptyline metabolism examined in hairs with CYP2C19 and CYP2D6 polymorphisms. Int. J. Legal. Med. 2008, 122, 149–155

Koren, G.; Cairns, J.; Chitayat, D.; Gaedigk, A.; Leeder, S.J. Pharmacogenetics of morphine poisoning in a breastfed neonate of a codeine-prescribed mother. Lancet 2006, 368, 704.

10.3390/diagnostics11081307

Author Response

Reviewer 3

COMMENT: The manuscript is an interesting article on deaths related to the use of ethanol and psychotropic substances. The article is very interesting however there are some points that need to be clarified in the methods, results, and discussion, in order to proceed with the acceptance of the article which today appears confusing.

ANSWER: We are grateful for the kind words of the reviewer. As we answer to the first and second reviewers, there are a few studies of this type carried out in the European population. Having into account that it is not a minor issue that co-consumption of alcohol and medical psychotropics could modify the levels of ethanol in blood, especially in men, the toxicokinetic interactions between these drugs need to be explored in depth.

COMMENTS:

Methods: what PMI (post-mortem interval) did the deceased have? Were there any subjects with putrefactive phenomena? Were the corpses kept in cells? This aspect is very important because in the stages of putrefaction, with autolysis, the increase of ethanol in the blood has been recorded and this could be an artifact.

ANSWER: As stated in the manuscript (Materials and Methods Section, Toxicological analyses, lines 97-99, “In general, the bodies were in a good state of preservation and kept refrigerated until the time of autopsy. Blood samples were stored at -7°C until analysis. All samples were analyzed less than 4 weeks after recovery”. Due to the fact that the post-mortem interval was not always available, we decided do not included this data in the study. Nevertheless, the bodies of decedents included in our study were in a good state of preservation and kept refrigerated until the time of autopsy, so there were not putrefactive phenomena, avoiding potential artifacts derived from putrefaction.

1) Methods: how was the toxicological examination done? Was the method unique or did it vary? What kind of method was used?

ANSWER: All the corpses were examined following standard forensic protocols. No specific protocols were used in any case. It should be noted that this work is a retrospective epidemiological work in which the toxicological results obtained during three years in standard autopsies are analyzed, and that is the great value of this work. As a consequence, our results reflect (without intervention or prior selection of the victims) the simultaneous consumption of alcohol and psychotropic drugs in an European population.

2) A different paragraph should be made for results and for discussion.

ANSWER: Although this suggestion of the referee may be very accurate, in our opinion (and based in the literature) this is the most appropriate way to show pharmacoepidemiological results. Thus, keeping the discussion immersed in the results, our results are fully understood by the reader. It would be very different in the case of a mechanistic article, developed in a laboratory for example, in which case I agree with the reviewer that it would be necessary to separate the Results and Discussion sections.

3) The results are confusing, they must be exposed uncritically and the tables in the text must be explained.

ANSWER: We agree with the reviewer in this criticism, which is why the Results and Discussion section has been modified and rewritten. In addition, and as answered to the editor previously, Table 3a has been transformed into a figure (Fig.1) to make the manuscript easier to understand for the reader.

4) Results: why were sudden deaths and gastrointestinal disorders inserted in line 156? What does that have to do with the study?

ANSWER: As mentioned above, this is a retrospective epidemiological study, so all types of deaths observed in the autopsies performed during the three years under study were included, even decedents who suffered sudden death or gastrointestinal disorders.

5) Discussion:

The authors should better explain the purpose of the study. Why should the scientific community be interested? Was the cause of death related to the study of these substances acutely or chronically?

ANSWER: As stated in the Title of the article the main purpose of the study is evaluate the relevance of ethanol and Medical Psychotropics co-consumption in European countries. Nowadays, there are only a few studies analyzing this simultaneous consumption of both type of psychotropics in Europe. And in our opinion, the best way to approach this objective is through a retrospective epidemiological study.

In no case was the cause of death related to the consumption (both, acutely or chronically) of such psychotropic drugs because the objective was to approach the prevalence, determinants, and consequences of this co-consumption in the European population.

Nevertheless, we have rewritten the Results and Discussion Section (lines 140-146) to clarify this point as follow:

“We conducted a retrospective observational study analyzing toxicological data from blood samples collected from 380 compulsory forensic autopsies of adult subjects performed during the years 2015 to 2017 at the Institute of Legal Medicine of Las Palmas (Spain). Our results and in-depth review of the literature on this topic provide important new insights on the prevalence, determinants, and potential interactions between alcohol and psychoactive medications, taken alone or in combination, in a European adult population involved in medico-legal issues.”

6) it is necessary to cite these studies in the present study:

https://doi.org/10.3390/toxics9110292

Wolf, C.R.; Smith, G.; Smith, R.L. Pharmacogenetics. BMJ 2000, 320, 987-990.

Thieme, D.; Rolf, B.; Sachs, H.; Schmid, D. Correlation of inter-individual variations of amitriptyline metabolism examined in hairs with CYP2C19 and CYP2D6 polymorphisms. Int J Legal Med 2008, 122, 149-155.

Koren, G.; Cairns, J.; Chitayat, D.; Gaedigk, A.; Leeder, S.J. Pharmacogenetics of morphine poisoning in a breastfed neonate of a codeine-prescribed mother. Lancet 2006, 368, 704.

ANSWER: Following the appropriate suggestion of the referees the authors have read and studied in detail the above cited references and some of them have been included in the text and in the references section. It has to be taken into account that the relevance of pharmacogenetics and the different metabolizer phenotypes is only tangential in this article, and is not a primary objective of our work, so the reviewer has to understand that all the suggested references cannot be introduced in the new version of the article.

Consequently, as suggested by the referee, in line 76 (Introduction Section) has been introduced the new reference [16]; (Di Nunno, N.; Esposito, M.; Argo, A.; Salerno, M.; Sessa F. Pharmacogenetics and Forensic Toxicology: A New Step towards a Multidisciplinary Approach. Toxics 2021, 9, 292).

And similarly, line in 253 (Results and Discussion Section) has been introduced another new reference [39]; (Wolf, C.R.; Smith, G.; Smith, R.L. Pharmacogenetics. BMJ 2000, 320, 987-990).

However, in our opinion, the rest of references (Thieme et al. and Koren et al.), must not to be included in this article.

Reviewer 4 Report

Ethanol and Medical Psychotropics co-consumption in European Countries. Results from a Three-Year Retrospective Study  in Forensic samples in Spain.

Article covers a relevant field of interest of forensic toxicology on the point of view of epidemiology  research.   

Within Authors conclusions: "Our results demonstrate that legal psychotropics are frequently found in compulsory autopsies of adult subjects in Spain (see Table 2). These findings agree with those previously published, which indicate that BZDs, opiates, and ethanol are the most frequently detected substances in forensic toxicological analyses in European [8-10] and non-European countries [4,17]."

Please consider in this regard: Toxicological Findings of Self-Poisoning Suicidal Deaths: A Systematic Review by Countries ( Albano, Giuseppe Davide et. al, ). Toxics2022, 10(11), 654.

Conclusions:" Our results showed a high prevalence of co-consumption of MPs and  ethanol in the Spanish population involved in medico-legal issues and suggest that toxicokinetic  interactions may be lowering BAC in men. This is a very worrying result as it could indicate that  the legal blood ethanol limits set by the legislation would not be appropriate for men on MPs treatment".

As the main object is related to retrospective analysis of forensic autopsies during the period 2015.2017 in Canary Islands, the main focus is no coherent with the title of article (Ethanol and Medical Psychotropics co-consumption in European Countries) as indicated by authors.

 Authors would specify relevant novelty in the present analysis in regard to their previous paper (ref. n. 19). 

Author Response

Reviewer 4

COMMENT: Ethanol and Medical Psychotropics co-consumption in European Countries. Results from a Three-Year Retrospective Study in Forensic samples in Spain. Article covers a relevant field of interest of forensic toxicology on the point of view of epidemiology research.

ANSWER: We are grateful for the kind words of the reviewer. As we answer to the first and second reviewers, there are a few studies of this type carried out in the European population. Having into account that the fact that co-consumption of alcohol and medical psychotropics may modify the levels of ethanol in blood, especially in men, new and more extensive epidemiological studies should be developed in European populations.

COMMENT: Within Authors conclusions: “Our results demonstrate that legal psychotropics are frequently found in compulsory autopsies of adult subjects in Spain (see Table 2). These findings agree with those previously published, which indicate that BZDs, opiates, and ethanol are the most frequently detected substances in forensic toxicological analyses in European [8-10] and non-European countries [4,17].”

Please consider in this regard: Toxicological Findings of Self-Poisoning Suicidal Deaths: A Systematic Review by Countries (Albano, Giuseppe Davide et. al, Toxics, 2022, 10(11), 654).

ANSWER: Following the appropriate suggestion of the referee, the reference by Albano et al. (2022), has been introduced in the line Results and Discussion Section (line 294) and numbered with the number [43], and this change has also been performed in the line 400 in the References Section. This reinforces the message that the misuse of PMs is a very serious problem in Western populations, as this article literally states that “In Europe and Western countries was observed higher rates of illicit drugs (especially opioids) and medically prescribed drugs (especially benzodiazepines, antidepressants, and neuroleptics) for suicide purposes”.

COMMENT: Conclusions: "Our results showed a high prevalence of co-consumption of MPs and ethanol in the Spanish population involved in medico-legal issues and suggest that toxicokinetic interactions may be lowering BAC in men. This is a very worrying result as it could indicate that the legal blood ethanol limits set by the legislation would not be appropriate for men on MPs treatment".

ANSWER: The reviewer is right on this point, since the text should be modified in the sense of the title and, therefore, we should speak of the European population and not the Spanish population.

Such a change has been introduced in the new version of the manuscript in the Abstract Section (line 33).

COMMENT: As the main object is related to retrospective analysis of forensic autopsies during the period 2015.2017 in Canary Islands, the main focus is no coherent with the title of article (Ethanol and Medical Psychotropics co-consumption in European Countries) as indicated by authors.

ANSWER: Unfortunately, we do not agree with the reviewer on this point. In our opinion, the title is fully consistent with the design of the study (retrospective epidemiological study in a European population).

COMMENT: Authors would specify relevant novelty in the present analysis in regard to their previous paper (ref. n. 19).

ANSWER: The novelty of the present work with respect to previous studies from our group, we believe, is evident from the title to the conclusions It is clear that in Europe a serious problem exists with the high simultaneous consumption of alcohol and medical psychotropics. Previous studies reinforce and give an idea of the real magnitude we have in Europe with the simultaneous consumption of alcohol and medical psychotropics.

Round 2

Reviewer 3 Report

The article was sufficiently improved and the authors provided correct answers to all comments made by the reviewer. I think it's ready for publication.

Reviewer 4 Report

Authors  considered all suggestions, by substantial review of the article. Points of criticism were clarified.